# Multi-agent Optimistic Soft Q-Learning: A co-MARL Algorithm with a Global Convergence Guarantee

## Abstract

Despite the empirical success of cooperative multi-agent reinforcement learning algorithms in recent years, the theoretical understandings, especially for algorithms under the centralized training with decentralized execution (CTDE) framework, are still lacking. Interestingly, existing algorithms sometimes fail to handle some seemingly simple tasks. Motivated by these failed cases, this paper proposes multi-agent optimistic soft Q-learning (MAOSQL), a new co-MARL algorithm with a global convergence guarantee. The design of MAOSQL includes an optimistic local Q-function and a softmax local policy, which naturally leads to a different objective from existing algorithms. We show that optimizing this objective gives near-optimal policies with a tractable error bound, and MAOSQL provably converges to the global optima with properly chosen hyper-parameters. Further, we show that MAOSQL can be easily modified for deep reinforcement learning, MAOSDQN. We evaluate MAOSDQN in didactic environments where value decomposition methods or policy gradient methods fail, as well as level-based foraging, a popular MARL benchmark. The results confirm our theoretical analysis and indicate the potential of our proposed method to deal with more complicated problems.

## 1 Introduction

Cooperative multi-agent reinforcement learning (co-MARL) deals with systems in which multiple agents collectively learn local policies to maximize a global objective by interacting within a common environment. There are many real-world applications of co-MARL, including traffic light systems (Wiering et al. (2000)), power networks (Wang et al. (2021)), autonomous vehicles (Cortes et al. (2004)), video games (Vinyals et al. (2019)), etc.

A popular framework for solving co-MARL problems is centralized training with decentralized execution (CTDE) (Kraemer & Banerjee (2016)). This framework exploits the fact that a co-MARL problem can be converted to a single-agent RL problem by learning the optimal policy at a central entity while restricting the learned policy to be distributed such that agents can make independent decisions during execution. Following this framework, both value decomposition methods (Sunehag et al. (2017); Rashid et al. (2018)) and policy gradient methods (Foerster et al. (2018); Yu et al. (2022)) have had empirical success in recent years. However, these algorithms fail to converge to a global optimal optimum in some simple environments because of miscoordination and relative overgeneralization (Wei & Luke (2016)).

In this paper, we develop a co-MARL algorithm with global convergence guarantees by overcoming miscoordination and relative overgeneration. We show that each MARL algorithm defines a projection from the joint action space to each local action space, which can be specified by defining local value functions and the relations between local value functions and local policies. Based on this observation, we propose multi-agent optimistic soft Q-learning (MAOSQL) by defining such a projection. Its local value function is optimistic about cooperating agents, and the way it determines local policy resembles that of soft Q-learning. This definition naturally leads to a different objective from existing algorithms, which can be viewed as an extension of traditional value functions. We

prove that optimizing this objective gives a near-optimal policy with a tractable error bound, and MAOSQL provably converges to a global optimum with properly chosen hyper-parameters.

Further, we show that MAOSQL can be easily modified for deep reinforcement learning, MAOS-DQN, by using a centralized critic network to approximate global Q-function and decentralized actor networks to maintain the approximated local Q-functions for each agent. We evaluate this actor-critic style algorithm in tensor game, matrix game and level-based foraging. The former two environments aim to show that MAOSDQN can avoid miscoordination and relative overgeneralization, respectively. Results in the last environment indicate the potential of our proposed method to deal with more complex problems.

## 2 PRELIMINARIES

### 2.1 COOPERATIVE MULTI-AGENT REINFORCEMENT LEARNING PROBLEM

In this paper, we consider the infinite-horizon fully observable cooperative multi-agent reinforcement learning, which can be modeled as a Markov decision process $G = \langle N, \mathcal{S}, \mathcal{A}, \mathcal{P}, \rho, r, \gamma \rangle$. Here, $N$ is the number of agents, $\mathcal{S}$ the state space is visible to all agents, $\mathcal{A} = \{\mathcal{A}^n\}_{n=1}^N$ is the action space where $\mathcal{A}^n$ is the action space of agent $n$, $\mathcal{P} : \mathcal{S} \times \mathcal{A} \to \Delta(\mathcal{S})$ is the transition function, $\rho \in \Delta(\mathcal{S})$ is the initial state distribution, $r : \mathcal{S} \times \mathcal{A} \to [0, 1]$ is the reward function shared by all agents, and $\gamma \in (0, 1)$ is the discount factor.

Given a policy $\pi : \mathcal{S} \to \Delta(\mathcal{A})$, the value function and action-value function under $\pi$ are defined as

$$V^\pi(s) = \mathop{\mathbb{E}}_{\substack{s_0=s, a_t \sim \pi(\cdot|s_t) \\ s_{t+1} \sim \mathcal{P}(\cdot|s_t, a_t)}} \left[ \sum_{t=0}^\infty \gamma^t r(s_t, a_t) \right], \ V^\pi(\rho) = \mathop{\mathbb{E}}_{s \sim \rho} [V^\pi(s)]$$

$$Q^\pi(s, a) = r(s, a) + \gamma \mathop{\mathbb{E}}_{s' \sim \mathcal{P}(\cdot|s, a)} [V^\pi(s')]$$

The goal of the agents is to optimize $V^\pi(\rho)$. Let $\pi^* = \arg\max_\pi V^\pi(\rho)$ and $V^* = V^{\pi^*}$. According to Bellman & Dreyfus (1959), $\pi^*$ simultaneously maximizes $V^\pi(s)$ for all $s \in \mathcal{S}$.

### 2.2 DISTRIBUTED POLICIES AND LOCAL VALUE FUNCTIONS

We say that a policy $\pi$ is a distributed policy (or a localized policy) if and only if there exists $\{\pi^n : \mathcal{S} \to \Delta(\mathcal{A}^n)\}_{n=1}^N$, such that $\pi(a|s) = \prod_{n=1}^N \pi^n(a^n|s), \forall s, a$, i.e. agents make decisions based on the state only, independently of other agents' decisions. In many MARL scenarios where there is no central controller, the joint policy must be distributed.

A co-MARL problem can be viewed as a single-agent RL problem, with the constraint that the policy has to be distributed. From a centralized point of view, MARL algorithms are projecting the single-agent RL problem onto each agent's point of view. Interestingly, for most existing methods, this projection can be decomposed into two steps. The first step is to calculate the local Q-function from the global Q-function, and the second step is to determine the local policy based on the local Q-function. Here are some examples. Directly applying vanilla Q-learning on MARL problems (Lauer & Riedmiller (2000)) by regarding all agents as one is equivalent to taking the maximum of the global Q-function over other agents' actions to be the local Q-function. Independent Q-learning (Tan (1993)), which regards other agents as part of the environment, is equivalent to taking the expectation of the global Q-function over other agents' behavior policy to be the local Q-function. Value decomposition methods (Sunehag et al. (2017); Rashid et al. (2018; 2020); Son et al. (2019); Wang et al. (2020a)) are learning an approximated projection of the Q-function that satisfies the Individual-Global-Max principle (Son et al. (2019)). All the algorithms above adopt a greedy policy according to the local Q-functions. Policy gradient methods (Lowe et al. (2017); Foerster et al. (2018); Yu et al. (2022)) perform gradient updates on policy parameters according to the policy gradient theorem (Sutton et al. (1999)). The gradients can be written in terms of the expectation of the global Q-function over other agents' current policy. See appendix A for a detailed derivation.

| Algorithm | Local Q-function | Local Policy |
|---|---|---|
| **QL** | $Q^n(s,a^n) = \max_{a^{-n}} Q(s,a)$ | $\pi^n(\cdot|s) = \arg\max_{a^n} Q^n(s,a^n)$ |
| **IQL** | $Q^n(s,a^n) = \mathbb{E}_{a^{-n}\sim\pi_b^{-n}} Q(s,a)$ | $\pi^n(\cdot|s) = \arg\max_{a^n} Q^n(s,a^n)$ |
| **VD** | $Q(s,a) = \text{MIX}\left(\{Q^n(s,a^n)\}; s,a\right)$ | $\pi^n(\cdot|s) = \arg\max_{a^n} Q^n(s,a^n)$ |
| **PG** | $Q^{\pi_\theta,n}(s,a^n) = \mathbb{E}_{a^{-n}\sim\pi_\theta^{-n}} Q^{\pi_\theta}(s,a)$ | $\frac{\partial V^{\pi_\theta}(\rho)}{\partial \pi_\theta^n(a^n|s)} = \frac{1}{1-\gamma} d_\rho^{\pi_\theta}(s) Q^{\pi_\theta,n}(s,a_n)$ |
| **MAOSQL** | $Q^n(s,a^n) = \frac{1}{\beta}\log\mathbb{E}_{a^{-n}\sim\zeta^{-n}} e^{\beta Q(s,a)}$ | $\pi^n(a^n|s) = \frac{e^{\alpha Q^n(s,a^n)}}{\sum_{a'^n} e^{\alpha Q^n(s,a'^n)}}$ |

Table 1: Summary of the definition of local value functions and relation between policy and local value functions in some MARL algorithms

## 2.3 Two Issues of a Distributed Policy

The set of distributed policies is a very small subset of the set of all joint policies. For example, consider a system with $N$ agents and each agent has $m$ actions to choose from, then the DoF (degree of freedom) of the joint policy class is $m^N - 1$, while it is only $N(m-1)$ when being restricted to distributed policies. Therefore, the projection from global to local may lead to the loss of optimality, resulting in failures of existing algorithms on seemingly simple tasks. It has been observed that this failure is due to the following two issues (Wei & Luke (2016)): miscoordination and relative overgeneralization.

### 2.3.1 Miscoordination

The problem of miscoordination arises when the underlying MDP has symmetric actions, while the projection cannot break this symmetry. Consider a single-state 2-agent game where each agent has 2 actions (labeled as 1,2). Agents are awarded 1 if and only if they take different actions from each other's. For Q-learning and value decomposition methods, we have $Q^1(s,1) = Q^1(s,2), Q^2(s,1) = Q^2(s,2)$ at convergence by definition, so joint action pairs $(1,1),(2,2),(1,2),(2,1)$ are all optimal actions given by the local policy, but only the latter two are real optimal actions. Although for value-based methods, this problem can be alleviated by randomization when deep neural networks are involved, the problem persists in slightly more complicated environments (See Section 4.1).

### 2.3.2 Relative Overgeneralization

The problem of relative overgeneralization, also being referred to as centralized-decentralized mismatch (Wang et al. (2020b)), arises when the optimal joint action is risky in the sense that the return would be very low if some of the collaborating agents do not take the appropriate actions.

Consider a single-state 2-agent matrix game where each agent has 3 actions (labeled as 1,2,3). The reward matrix $R$ is as shown in Table 2, where $R_{ij}$ is the reward when agent 1 takes action $i$ and agent 2 takes action $j$. The optimal joint policy for the two agents is to perform action 3 simultaneously.

| $a^1$ \ $a^2$ | 1 | 2 | 3 |
|---|---|---|---|
| 1 | 2 | 2 | 0 |
| 2 | 2 | 2 | 0 |
| 3 | 0 | 0 | 3 |

Table 2: Rewards of the single-state 2-agent matrix game. Optimal action is highlighted.

Policy gradient methods always fail to converge to the globally optimal solution even with exact gradients and sufficient policy parameters. If we simulate a gradient ascend algorithm starting from a uniform initial policy, then we will see that $Q^1(s,1) = Q^1(s,2) = Q^2(s,1) = Q^2(s,2) = 4/3$, while $Q^1(s,3) = Q^2(s,3) = 1$. So the algorithm continues to decrease the probability of choosing action 3 and finally converges to the sub-optimal policy of choosing only action 1 and 2.

Among value decomposition methods, VDN (Sunehag et al. (2017)) and QMIX (Rashid et al. (2018)) fail in this case due to the limited expressiveness of the mixing function. Note that as long as

the maximum of the approximated global Q function matches that of the real one, the algorithm can give an optimal policy. Therefore, WQMIX (Rashid et al. (2020)) tries to distinguish the important joint actions and give them higher weights in the loss function, while other works like QTRAN (Son et al. (2019)) and QPLEX (Wang et al. (2020a)) try to extend the expressiveness. However, there is no theoretical guarantee that these methods can resolve this issue.

## 3 ALGORITHM DESIGN

### 3.1 MULTI-AGENT OPTIMISTIC SOFT VALUE FUNCTIONS

To address the two issues, we focus on designing a projection of the global Q-function (denoted as $\tilde{Q}$) onto local Q-function (denoted as $\hat{Q}$), together with a mapping from local Q-functions to distributed policy, such that (1) they can break the symmetry between actions and (2) they can distinguish the optimal joint action even if it is risky. Therefore, we propose to enforce $\hat{Q}$ to be the softmax of $\tilde{Q}$ over other agents and use the energy-based policy $\hat{\pi}$ (Haarnoja et al. (2017)).

$$\hat{Q}_{\zeta}^n(s, a^n) = \frac{1}{\beta} \log \mathbb{E}_{a^{-n} \sim \zeta^{-n}} \left[ e^{\beta \tilde{Q}(s,a)} \right]$$

$$\hat{\pi}_{\hat{Q}}^n(a^n|s) = \frac{e^{\alpha \hat{Q}^n(s,a^n)}}{\sum_{a'^n} e^{\alpha \hat{Q}^n(s,a'^n)}}$$

Here $\alpha, \beta \in \mathbb{R}_+$ are hyper-parameters and $\zeta$ is an arbitrary policy such that $\zeta(a|s) > 0, \forall s, a$. Intuitively, $\zeta$ can break the symmetry between actions, and $\beta$ indicates the optimistic level of an agent about other agents. A large $\beta$ encourages agents to choose high-return yet risky actions.

The following lemma shows that the given policy is optimal in the sense that it maximizes a regularized expected return.

**Lemma 1.** *Define*

$$J_{\hat{Q}(s,\cdot)}(\pi) = \frac{1}{N} \sum_{n=1}^{N} \mathbb{E}_{a^n \sim \pi^n} \left[ \hat{Q}^n(s, a^n) - \frac{1}{\alpha} \log \pi^n(a^n|s) \right]$$

*then $\hat{\pi}_{\hat{Q}}^{1:N}$ maximizes $J_{\hat{Q}(s,\cdot)}(\pi)$ for all $s \in \mathcal{S}$.*

The proof of the lemma can be found in Appendix B.

Note that $J_{\hat{Q}(s,\cdot)}(\pi)$ can be viewed as an evaluation of state-value with an entropy regularization term

$$\hat{V}_{\zeta}^{\pi,n}(s) = \mathbb{E}_{a^n \sim \pi^n} \left[ \hat{Q}_{\zeta}^n(s, a^n) - \frac{1}{\alpha} \log \pi^n(a^n|s) \right]$$

$$\tilde{V}_{\zeta}^{\pi}(s) = \frac{1}{N} \sum_{n=1}^{N} \hat{V}_{\zeta}^{\pi,n}(s)$$

so given $\pi$ and $\zeta$, we can define a Bellman operator $\mathcal{T}_{\alpha,\beta,\pi,\zeta}$ on $\tilde{Q}$. When it causes no confusion, we denote it as $\mathcal{T}_{\pi,\zeta}$ for simplicity.

$$\mathcal{T}_{\pi,\zeta}\tilde{Q}(s,a) = r(s,a) + \gamma \mathbb{E}_{s' \sim P(\cdot|s,a)}[\tilde{V}_{\zeta}^{\pi}(s')]$$

**Lemma 2.** *$\mathcal{T}_{\pi,\zeta}$ is a $\gamma$-contraction.*

According to **Lemma 2** we can formally define the multi-agent optimistic soft (action-)value functions and optimal policies:

$$\tilde{Q}_{(\alpha,\beta,\zeta)}^{\pi} \text{ uniquely solves } \mathcal{T}_{\alpha,\beta,\pi,\zeta}\tilde{Q} = \tilde{Q}$$

$$\pi_{(\alpha,\beta,\zeta)}^* = \arg\max_{\pi} \tilde{Q}_{(\alpha,\beta,\zeta)}^{\pi}$$

$$\tilde{Q}_{(\alpha,\beta,\zeta)}^* = \tilde{Q}_{(\alpha,\beta,\zeta)}^{\pi_{(\alpha,\beta,\zeta)}^*}$$

According to **Lemma 1**, $\pi^*_{(\alpha,\beta,\zeta)} = \hat{\pi}_{\hat{Q}^*_{(\alpha,\beta,\zeta)}}$ where $\hat{Q}^{*n}_{(\alpha,\beta,\zeta)} = \frac{1}{\beta} \log \mathbb{E}_{a^{-n} \sim \zeta^{-n}} \left[ e^{\beta \tilde{Q}^*_{(\alpha,\beta,\zeta)}(s,a)} \right]$.

As $\alpha$ approaches infinity, the entropy regularization term approaches 0 and hence the policy approaches a deterministic one. In this case, we use the subscript $(+\infty, \beta, \zeta)$. If $\beta$ approaches infinity, then softmax goes to max and $\zeta$ has no impact, so we simplify the subscript to be $(\alpha, +\infty)$ in this case. If $\alpha$ and $\beta$ both approach infinity, then this definition is just equivalent to the original definition of value functions (subscripted by $(+\infty, +\infty)$), so it can be viewed as an extension of value functions.

## 3.2 Optimizing the Optimistic Soft Value Functions

---

**Algorithm 1** Multi-Agent Optimistic Soft Value Iteration (MAOSVI)

---

Initialize: $\tilde{Q}_0 : \mathcal{S} \times \mathcal{A} \to \left[0, \frac{1}{1-\gamma}\right], \zeta^{1:N} : \mathcal{S} \to \Delta\left(\mathcal{A}^{1:N}\right)$
**for** $t = 0, 1, \cdots, T-1$ **do**
$\quad \hat{Q}^n_t(s, a^n) \leftarrow \frac{1}{\beta} \log \mathbb{E}_{a^{-n} \sim \zeta^{-n}} \left[ e^{\beta \tilde{Q}_t(s,a)} \right]$
$\quad \hat{V}^n_t(s) \leftarrow \frac{1}{\alpha} \log \sum_{a^n} e^{\alpha \hat{Q}^n_t(s, a^n)}$
$\quad \tilde{V}_t(s) \leftarrow \frac{1}{N} \sum_{n=1}^N \hat{V}^n_t(s)$
$\quad \tilde{Q}_{t+1}(s, a) \leftarrow r(s, a) + \gamma \mathbb{E}_{s' \sim P(\cdot|s,a)}[\tilde{V}_t(s')]$
**end for**

---

To find the optimal policy $\pi^*_{(\alpha,\beta,\zeta)}$, or equivalently, the optimal Q-function $\tilde{Q}^*_{(\alpha,\beta,\zeta)}$, a straightforward idea is to alternate between policy update and policy evaluation. In particular, if we perform exactly one evaluation step between two policy update steps, then in the evaluation step we have $\pi = \hat{\pi}_{\hat{Q}}$, which implies

$$\hat{V}^{\pi,n}_\zeta(s) = \mathbb{E}_{a^n \sim \hat{\pi}^n} \left[ \hat{Q}^n(s, a^n) - \frac{1}{\alpha} \log \hat{\pi}^n(a^n|s) \right]$$

$$= \mathbb{E}_{a^n \sim \hat{\pi}^n} \left[ \hat{Q}^n(s, a^n) - \frac{1}{\alpha} \left( \alpha \hat{Q}^n(s, a^n) - \log \sum_{a'^n} e^{\alpha \hat{Q}^n(s, a'^n)} \right) \right]$$

$$= \frac{1}{\alpha} \log \sum_{a^n} e^{\alpha \hat{Q}^n(s, a^n)}$$

Therefore, we obtain **Algorithm 1** that resembles Value Iteration Algorithm.

**Theorem 1.** *(Convergence of MAOSVI)* $\tilde{Q}_T$ converges to $\tilde{Q}^*$ as $T \to \infty$.

The following theorem shows that optimizing the optimistic soft value functions gives a policy that is close to the optimal policy of the original problem.

**Theorem 2.** *(Near Optimality of Optimistic Soft Policies)*

1. *Assume that $\pi^*_{(+\infty,\beta,\zeta)}$ is unique, and let $z = \min_{s \in \mathcal{S}} \zeta\left(a^*_{(+\infty,\beta,\zeta)}(s) \mid s\right)$, then*

$$\left\| \tilde{Q}^*_{(+\infty,+\infty)} - \tilde{Q}^*_{(\alpha,\beta,\zeta)} \right\|_\infty \leq \left( \frac{\log|\mathcal{A}|}{N\alpha} + \frac{\log\frac{1}{z}}{\beta} \right) \frac{\gamma}{1-\gamma} = O\left( \frac{1}{\alpha} + \frac{\log\frac{1}{z}}{\beta} \right).$$

2. *Let $\Delta = \min_{s \in \mathcal{S}} \left( \tilde{Q}^*_{(+\infty,+\infty)}(s, a^*_{(+\infty,+\infty)}(s)) - \max_{a \notin a^*_{(+\infty,+\infty)}(s)} \tilde{Q}^*_{(+\infty,+\infty)}(s,a) \right)$,*

*and $\Delta' = \min_{s \in \mathcal{S}} \left( \tilde{Q}^*_{(+\infty,\beta,\zeta)}(s, a^*_{(+\infty,\beta,\zeta)}(s)) - \max_{a \neq a^*_{(+\infty,\beta,\zeta)}(s)} \tilde{Q}^*_{(+\infty,\beta,\zeta)}(s,a) \right)$.*

*If $\alpha > \frac{2\gamma \log|\mathcal{A}|}{(1-\gamma)\Delta'}, \beta > \frac{(1+\gamma)\log\frac{1}{z}}{(1-\gamma)\Delta}$, then we have $\left\| \tilde{Q}^{\pi^*_{(\alpha,\beta,\zeta)}}_{(+\infty,+\infty)} - \tilde{Q}^*_{(+\infty,+\infty)} \right\|_\infty \leq$*

$$\left( \frac{|\mathcal{A}|^{\frac{2\gamma}{(1-\gamma)N}} \sum_n |\mathcal{A}^n|}{(1-\gamma)e^{\alpha\Delta'}} + \frac{2\log|\mathcal{A}|}{N\alpha} + \frac{\log\frac{1}{z}}{\beta} \right) \frac{\gamma}{1-\gamma} = O\left( \frac{1}{\alpha} + \frac{\log\frac{1}{z}}{\beta} \right).$$

Note that we only assume the uniqueness of $\pi^*_{(+\infty,\beta,\zeta)}$ but not the uniqueness of $\pi^*_{(+\infty,+\infty)}$. This assumption is realistic because we can break the symmetry between actions by assigning different $\zeta(\cdot|s)$ on symmetric actions.

### 3.3 TRAINING DEEP Q-NETWORKS

---

**Algorithm 2** Multi-Agent Optimistic Soft Q-Learning (MAOSQL)

---

Initialize: $\hat{Q}_0^{1:N} : \mathcal{S} \times \mathcal{A}^{1:N} \to \left[0, \frac{1}{1-\gamma}\right]$
**for** $t = 0, 1, \cdots, T-1$ **do**
   $\hat{V}_t^n \leftarrow \frac{1}{\alpha} \log \sum_{a^n} e^{\alpha \hat{Q}_t^n(s,a^n)}$
   $\hat{\pi}_t^n(a^n|s) \leftarrow \exp\left(\alpha\left(\hat{Q}_t^n(s,a^n) - \hat{V}_t^n(s)\right)\right)$
   $\hat{\zeta}_t(a|s) \leftarrow \frac{\epsilon}{|\mathcal{A}|} + (1-\epsilon)\hat{\pi}_t(a|s)$
   $\tilde{Q}_{t+1}(s,a) \leftarrow r(s,a) + \gamma \mathbb{E}_{s'}\left[\frac{1}{N}\sum_{n'=1}^N \hat{V}_t^{n'}(s')\right]$
   $\hat{Q}_{t+1}^n(s,a^n) \leftarrow \frac{1}{\beta} \log \mathbb{E}_{a^{-n} \sim \hat{\zeta}_t^{-n}}\left[e^{\beta \tilde{Q}_{t+1}(s,a)}\right]$
**end for**
$\hat{V}_T^n \leftarrow \frac{1}{\alpha} \log \sum_{a^n} e^{\alpha \hat{Q}_T^n(s,a^n)}$
$\hat{\pi}_T^n(a^n|s) \leftarrow \exp\left(\alpha\left(\hat{Q}_T^n(s,a^n) - \hat{V}_T^n(s)\right)\right)$

---

**Algorithm 3** Multi-Agent Optimistic Soft Deep Q-Learning (MAOSDQN)

---

Initialize: $\theta, \psi, \alpha, \beta$
$\bar{\psi} \leftarrow \psi$
$D \leftarrow \emptyset$
**for** $\tau = 0, 1, 2, \cdots$ **do**
   Initialize $s_0 \sim \rho(\cdot)$
   **for** $t = 0, 1, 2, \cdots, T-1$ **do**
      Take action $a_t^n$ according to some behavior policy
      Observe $r_t, s_{t+1}$
      $D \leftarrow D \bigcup \{(s_t, a_t, r_t, s_{t+1})\}$
      w.p. $1 - \gamma$, sample $s_{t+1} \sim \rho(\cdot)$
   **end for**
   **for** each gradient step **do**
      $\theta \leftarrow \theta - \eta_\theta \nabla_\theta L_{\tilde{Q}}(\theta)$
      $\psi \leftarrow \psi - \eta_\psi \left(\nabla_\psi L_{\hat{Q}}(\psi) + \nabla_\psi L_{extra}(\psi)\right)$
      $\bar{\psi} \leftarrow \bar{\eta}\psi + (1 - \bar{\eta})\bar{\psi}$
   **end for**
**end for**

---

In this section, we extend the algorithm with deep neural networks. The extension uses a neural network to approximate the global Q table and our method allows us to put the joint action to the input of the neural network, while vanilla DQN requires the output dimension of the neural network to be the size of the joint action space. According to **Theorem 2**, choosing a $\zeta$ that is similar to the real optimal policy will give a better result by increasing $z$, and hence reducing the $\frac{\log \frac{1}{z}}{\beta}$ term. Therefore, we propose to use epsilon-decentralized-softmax policy, i.e. $\hat{\zeta}_t(a|s) = \frac{\epsilon}{|\mathcal{A}|} + (1-\epsilon)\hat{\pi}_t(a|s)$, which upper-bounds that term by ensuring at least some probability of choosing any actions, and gradually reduces it during the training process according to the current policy. We also show that using this epsilon-decentralized-softmax $\zeta$ instead of a fixed one preserves the convergence guarantee by slightly strengthening the condition on $\alpha$ and $\beta$ (**Algorithm 2** and **Theorem 3**). Note that $\hat{\zeta}_t(\cdot|s)$ can be a non-distributed policy, but we can still abuse the notation of $\hat{\zeta}_t^{-n}(a^{-n}|s)$ by defining $\hat{\zeta}_t^{-n}(a^{-n}|s) = \sum_{a^n} \hat{\zeta}_t(a|s)$.

**Theorem 3.** *(Convergence of MAOSQL) If $\beta > \frac{2(N-1)\alpha}{1-\gamma}$, then $\hat{\pi}_T^{1:N}$ converges to $\pi_*^{1:N}$ as $T \to \infty$, such that $\pi_*^{1:N}$ optimizes $\tilde{Q}^{\pi_*}$, the unique solution of Bellman equation $\mathcal{T}_{\pi,\zeta}\tilde{Q} = \tilde{Q}$ for some $\zeta$.*

In **Algorithm 2**, $\hat{Q}^n(s, \cdot)$ serves as the softmax-parameters of the policy, and $\tilde{V}^n(s)$ serves as the partition function. Hence, if the action space is discrete, we can approximate $\tilde{Q}$ and $\hat{Q}$ with neural networks, say $\tilde{Q}_\theta$ and $\hat{Q}_\psi$, respectively, which gives an actor-critic-style algorithm(**Algorithm 3**), Multi-Agent Optimistic Soft Deep Q-Learning (MAOSDQN).

We use TD-loss to train $\tilde{Q}_\theta$:

$$L_{\tilde{Q}}(\theta) = \mathbb{E}_{(s,a,r,s')\sim D} \left[ \frac{1}{2} \left( \tilde{Q}_\theta(s,a) - r - \gamma \frac{1}{N} \sum_{n=1}^N \frac{1}{\alpha} \log \sum_{a^n} e^{\alpha \hat{Q}_{\bar{\psi}}^n(s,a^n)} \right)^2 \right]$$

and the following loss to train $\hat{Q}_\psi$:

$$L_{\hat{Q}}(\psi) = \mathbb{E}_{\substack{s\sim D \\ a\sim \zeta_{\bar{\psi}}}} \left[ \frac{1}{N} \sum_{n=1}^N \frac{\left( e^{\beta \hat{Q}_\psi^n(s,a^n)} - e^{\beta \tilde{Q}_\theta(s,a)} \right)^2}{2\beta sg \left( e^{\beta \hat{Q}_\psi^n(s,a^n)} \right)} \right]$$

which ensures that

$$\nabla_\psi L_{\hat{Q}}(\psi) = \mathbb{E}_{\substack{s\sim D \\ a\sim \zeta_{\bar{\psi}}}} \left[ \frac{1}{N} \sum_{n=1}^N \left( e^{\beta \hat{Q}_\psi^n(s,a^n)} - e^{\beta \tilde{Q}_\theta(s,a)} \right) \nabla_\psi \hat{Q}_\psi^n(s,a^n) \right]$$

Here $D$ is the replay buffer, $\bar{\psi}$ is the target network, $\zeta_{\bar{\psi}}$ denotes the epsilon-softmax policy induced by $\hat{Q}_{\bar{\psi}}$, and $sg(\cdot)$ means stop gradient.

Empirically, to avoid overestimation on $\hat{Q}$, we need an extra loss for $\psi$:

$$L_{extra}(\psi) = \left[ \max \left( \hat{Q}_\psi^n(s,a^{*n}) - \tilde{Q}_\theta(s,a^*), 0 \right) \right]^2$$

where $a^* = \arg\max_a \tilde{Q}(s,a^*)$ is composed of $a^{*n} = \arg\max_{a^n} \hat{Q}^n(s,a^n)$.

## 4 EXPERIMENTS

In this section, we use numerical experiments to show that our proposed algorithms can resolve the two issues, miscoordination and relative overgeneralization, as well as the empirical performance of MAOSDQN on more complicated environments. We choose sota MAPG methods: MAA2C (Papoudakis et al. (2020)) and MAPPO (Yu et al. (2022)), and the popular value-based method QMIX (Rashid et al. (2018)) as baselines for comparison.

### 4.1 TENSOR GAME

This is a single-state game with $n$ agents, each has $m$ available actions and only one of them is the target actions. Rewards depend on the number of agents that take the target actions. In particular, we set $n = m = 3$ and the rewards are as shown in Table 3. To avoid an infinite loop, we set the horizon to be 10.

In this game, the maximum reward is achieved when there are exactly two of the agents taking the target action, hence there are 3 symmetric optimal joint actions. We use this example to demonstrate the miscoordinate issue.

| Number of agents taking the target action | 0 | 1 | 2 | 3 |
|---|---|---|---|---|
| Rewards | 0.6 | 0.0 | 1.0 | 0.5 |

Table 3: Rewards of the Tensor Game

As shown in Figure 1, MAOSDQN and policy gradient methods resolve the miscoordination issue, but QMIX fails and falls into a suboptimal policy. This result justifies our analysis in Section 2.3.1.

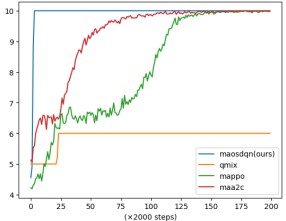 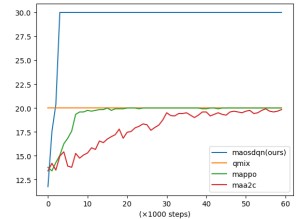 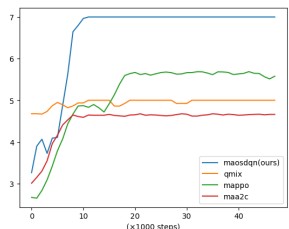

Figure 1: Training curves in Tensor Game

Figure 2: Training curves in single-state Matrix Game

Figure 3: Training curves in 3-state Matrix Game

| $s_1$ | | | | $s_2$ | | | | $s_3$ | | | |
|---|---|---|---|---|---|---|---|---|---|---|---|
| $a^1$ \ $a^2$ | 1 | 2 | 3 | $a^1$ \ $a^2$ | 1 | 2 | 3 | $a^1$ \ $a^2$ | 1 | 2 | 3 |
| 1 | 0.6 | 0 | 0 | 1 | 0.6 | 0 | 0.6 | 1 | 0.3 | 0.3 | 0 |
| 2 | 0 | 0.4 | 0.4 | 2 | 0 | 1.0 | 0 | 2 | 0.3 | 0.3 | 0 |
| 3 | 0 | 0.4 | 0.4 | 3 | 0.6 | 0 | 0.6 | 3 | 0 | 0 | 0.4 |

Table 4: Rewards of the 3-state 2-agent matrix game. Optimal actions are highlighted.

| $s_1$ | | | | $s_2$ | | | | $s_3$ | | | |
|---|---|---|---|---|---|---|---|---|---|---|---|
| $a^1$ \ $a^2$ | 1 | 2 | 3 | $a^1$ \ $a^2$ | 1 | 2 | 3 | $a^1$ \ $a^2$ | 1 | 2 | 3 |
| 1 | $s_1$ | $s_1$ | $s_1$ | 1 | $s_1$ | $s_2$ | $s_1$ | 1 | $s_1$ | $s_1$ | $s_3$ |
| 2 | $s_1$ | $s_3$ | $s_2$ | 2 | $s_2$ | $s_3$ | $s_2$ | 2 | $s_1$ | $s_1$ | $s_3$ |
| 3 | $s_1$ | $s_2$ | $s_3$ | 3 | $s_1$ | $s_2$ | $s_1$ | 3 | $s_3$ | $s_3$ | $s_2$ |

Table 5: Transition of the 3-state 2-agent matrix game. Optimal actions are highlighted.

## 4.2 MATRIX GAME

We consider a single-state version (shown in Table 2) and a 3-state version (shown in Table 4 and Table 5). In the 3-state version, the state is initialized randomly with uniform probability. To avoid an infinite loop, we set the horizon to be 10.

As shown in Figure 2 and Figure 3, only MAOSDQN converges to the optimal policy, while other algorithms fall into local optima. These results justify our analysis in Section 2.3.2.

## 4.3 LEVEL-BASED FORAGING

| Test Return Means \ Algorithm 
 Task | MAA2C | MAPPO | QMIX | MAOSDQN(ours) |
|---|---|---|---|---|
| 5x5-2p-1f-c | 0.738 | 0.840 | **1.000** | **1.000** |
| 8x8-2p-2f-c | 0.919 | 0.837 | **1.000** | **1.000** |
| 10x10-3p-5f | **0.985** | 0.945 | 0.407 | 0.168 |
| 10x10-3p-1f-pe | 0.535 | **0.639** | 0.518 | 0.492 |
| 12x12-6p-1f-pe | **0.780** | 0.726 | 0.020 | 0.331 |

Table 6: Level-based foraging tasks and results. Tasks are specified as "{height}x{width}-{num_agents}p-{num_foods}f(-c)(-pe)", where "c" means that any food can only be collected if all agents attempt to load it, "pe" means that agents are penalized if they try to load a food but failed.

In level-based Foraging (LBF) (Albrecht & Ramamoorthy (2015)), agents are required to collect food in a grid world. Agents and foods are assigned levels, such that a food item can be collected if and only if the sum of levels of the adjacent agents trying to collect it is greater or equal to the food's level. The full information of positions and levels of food and agents is visible to any agent. Each agent's action set includes moving in 4 directions and loading food. LBF allows a variety of

specific tasks with flexible map size, number of agents, number of foods, observability, cooperation setting, etc.

As shown in Table 6, MAOSDQN performs very well in relatively easier tasks. Its performance in harder tasks is still competitive.

## 5    RELATED WORK

The study of cooperative multi-agent reinforcement learning and its underlying structure, multi-agent Markov decision processes or team Markov games, dates back to the last century (Yoshikawa (1978); Ho (1980); Boutilier (1996)). This area has attracted great attention from the community in recent years due to advances in single-agent RL (Mnih et al. (2013; 2016); Schulman et al. (2017)). Conventional learning frameworks include independent learners and joint-action learners (Claus & Boutilier (1998)), but the former suffers from non-stationarity (Tan (1993)) while the latter does not apply to many scenarios due to partial observability or scalability issues. Hence, centralized training with decentralized execution (CTDE) has been widely adopted since its proposal (Kraemer & Banerjee (2016); Lowe et al. (2017)). Following this framework, there are mainly two series of algorithms, value decomposition methods (Sunehag et al. (2017); Rashid et al. (2018; 2020); Son et al. (2019); Wang et al. (2020a)) and policy gradient methods (Lowe et al. (2017); Foerster et al. (2018); Yu et al. (2022)), achieving competitive performance in practice.

Though empirically successful, unlike single-agent RL, the theoretical foundations of MARL, especially the CTDE framework, are relatively lacking. What's worse, the community seems to focus more on the convergence to Nash equilibrium in general-sum games (Hu & Wellman (1998; 2003); Littman (2001a;b)), leaving the results for cooperative settings as a corollary in special cases. However, as shown by Yongacoglu et al. (2021), a sub-optimal equilibrium can be arbitrarily worse than the optimal equilibrium. Here we only list some results for cooperative MARL. For joint action learners, Team Q-learning (Littman (2001b)) establishes the convergence to the optimal Q-function in general cases and the convergence to the optimal policy if the optima is assumed to be unique, optimal adaptive learning (OAL) (Wang & Sandholm (2002)) is the first algorithm with provable convergence to the optimal policy. For independent learners, distributed Q-learning (Lauer & Riedmiller (2000)) can be shown to converge in deterministic environments, Yongacoglu et al. (2021) provides an algorithm that converges to optimal equilibrium policy with high probability. Zhang et al. (2021) provides a more inclusive overview of MARL theories and algorithms.

The lack of theoretical foundations makes existing algorithms vulnerable to some specifically designed environments where miscoordination or relative overgeneralization is triggered. There are also empirical methods for dealing with these issues (Bowling & Veloso (2002); Matignon et al. (2007); Wei & Luke (2016); Yang et al. (2020)). As miscoordination can be alleviated by randomness, deep RL researchers have paid more attention to relative overgeneralization recently (Wei et al. (2018); Wang et al. (2020b); Jiang & Amato (2021)). However, none of them provide rigorous proof of the effectiveness of their proposed methods in general cases.

The idea of using softmax to replace max in Bellman operators was previously used by Asadi & Littman (2017), Gan et al. (2021) and Pan et al. (2019). The idea of softmax policy, or equivalently, adding entropy regularization to the learning objective, has been studied and proved effective in single-agent RL both empirically and theoretically (Haarnoja et al. (2017; 2018a;b); Mei et al. (2020)).

## 6    CONCLUSIONS AND FUTURE WORK

This paper presented MAOSQL, a new co-MARL algorithm with a global convergence guarantee. MAOSQL addresses the challenges of miscoordination and relative overgeneralization by using softmax local Q-function and softmax local policy, which differ from most existing methods. We provide a rigorous proof of convergence and near-optimality of our proposed algorithm and learning objective. Empirical results of MAOSDQN justify our theoretical analysis and show the potential of this method to deal with more complicated problems. We leave the problem of how to further improve the performance of MAOSDQN in complex environments for future work.

ACKNOWLEDGMENTS

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

## A  MULTI-AGENT POLICY GRADIENTS

Given a policy $\pi$, the (discounted) state visible distribution is defined as

$$d_{s_0}^\pi(s) = (1 - \gamma) \sum_{t=0}^\infty \gamma^t Pr(s_t = s | s_0, \pi, \mathcal{P}), \ d_\rho^\pi(s) = \mathbb{E}_{s_0 \sim \rho} \left[ d_{s_0}^\pi(s) \right]$$

When the policy parmeterization is differentiable, policy gradient theorem Sutton et al. (1999) allows us to express the gradients of the value functions over parameters in terms of action-value functions and gradients of policy parameterization:

$$\frac{\partial V^{\pi_\theta}(\rho)}{\partial \theta} = \frac{1}{1 - \gamma} \mathbb{E}_{s \sim d_\rho^{\pi_\theta}} \left[ \sum_{a \in \mathcal{A}} \frac{\partial \pi_\theta(a|s)}{\partial \theta} Q^{\pi_\theta}(s, a) \right]$$

This theorem can be easily extended to multi-agent settings. Assume that agent $n$'s policy is parameterized by $\theta^n$, let $Q^{\pi,n}(s, a^n) = \mathbb{E}_{a^{-n} \sim \pi^{-n}}[Q^\pi(s, a)]$, then we have

$$
\begin{aligned}
\frac{\partial V^{\pi_\theta}(\rho)}{\partial \theta^n} &= \frac{1}{1 - \gamma} \mathop{\mathbb{E}}_{s \sim d_\rho^{\pi_\theta}} \left[ \sum_{a \in \mathcal{A}} \frac{\partial \pi_\theta(a|s)}{\partial \theta^n} Q^{\pi_\theta}(s, a) \right] \\
&= \frac{1}{1 - \gamma} \mathop{\mathbb{E}}_{s \sim d_\rho^{\pi_\theta}} \left[ \sum_{a^n \in \mathcal{A}^n} \frac{\partial \pi_{\theta_n}^n(a^n|s)}{\partial \theta^n} Q^{\pi_\theta,n}(s, a_n) \right]
\end{aligned}
$$

# B  OMITTED PROOFS IN SECTION 3

**Lemma 1.** *Define*

$$
J_{\hat{Q}(s,\cdot)}(\pi) = \frac{1}{N} \sum_{n=1}^N \mathbb{E}_{a^n \sim \pi^n} \left[ \hat{Q}^n(s, a^n) - \frac{1}{\alpha} \log \pi^n(a^n|s) \right]
$$

*then $\hat{\pi}_{\hat{Q}}^{1:N}$ maximizes $J_{\hat{Q}(s,\cdot)}(\pi)$ for all $s \in \mathcal{S}$.*

*Proof.* Construct the Lagrangian function as

$$
L_{\zeta,s}(\pi, \lambda) = J_{\hat{Q}(s,\cdot)}(\pi) + \sum_{n=1}^N \lambda_n \left( \sum_{a^n} \pi^n(a^n|s) - 1 \right)
$$
$$
\frac{\partial L_{\zeta,s}(\pi, \lambda)}{\partial \pi^n(a^n|s)} = \frac{1}{N} \left( \hat{Q}^n(s, a^n) - \frac{1}{\alpha} \log \pi^n(a^n|s) + \frac{1}{\alpha} \right) + \lambda_n = 0
$$
$$
\implies \pi^n(a^n|s) \propto \exp\left\{ \alpha \hat{Q}^n(s, a^n) \right\}
$$

Hence, $\hat{\pi}_{\hat{Q}}^{1:N}$ maximizes $J_{\hat{Q}(s,\cdot)}(\pi)$ for all $s \in \mathcal{S}$. $\qquad\square$

**Lemma 2.** *$\mathcal{T}_{\pi,\zeta}$ is a $\gamma$-contraction.*

*Proof.* Assume that $\left\| \tilde{Q}_1 - \tilde{Q}_2 \right\|_\infty = \epsilon_0$, so $\tilde{Q}_1(s, a) \le \tilde{Q}_2(s, a) + \epsilon_0, \forall s, a$.

$$
\begin{aligned}
\mathcal{T}_{\pi,\zeta}\tilde{Q}_1(s, a) &= r(s, a) + \gamma \mathop{\mathbb{E}}_{s' \sim P(\cdot|s,a)} \left[ \frac{1}{N} \sum_{n=1}^N \mathop{\mathbb{E}}_{a'^n \sim \pi^n} \left[ \frac{1}{\beta} \log \mathop{\mathbb{E}}_{a'^{-n} \sim \zeta^{-n}} \left[ e^{\beta \tilde{Q}_1(s', a')} \right] - \frac{1}{\alpha} \log \pi^n(a'^n|s') \right] \right] \\
&\le r(s, a) + \gamma \mathop{\mathbb{E}}_{s'} \left[ \frac{1}{N} \sum_{n=1}^N \mathop{\mathbb{E}}_{a'^n \sim \pi^n} \left[ \frac{1}{\beta} \log \mathop{\mathbb{E}}_{a'^{-n}} \left[ e^{\beta(\tilde{Q}_2(s', a') + \epsilon_0)} \right] - \frac{1}{\alpha} \log \pi^n(a'^n|s') \right] \right] \\
&= r(s, a) + \gamma \mathop{\mathbb{E}}_{s'} \left[ \frac{1}{N} \sum_{n=1}^N \mathop{\mathbb{E}}_{a'^n \sim \pi^n} \left[ \frac{1}{\beta} \log e^{\beta \epsilon_0} \mathop{\mathbb{E}}_{a'^{-n}} \left[ e^{\beta \tilde{Q}_2(s', a')} \right] - \frac{1}{\alpha} \log \pi^n(a'^n|s') \right] \right] \\
&= r(s, a) + \gamma \mathop{\mathbb{E}}_{s'} \left[ \frac{1}{N} \sum_{n=1}^N \mathop{\mathbb{E}}_{a'^n \sim \pi^n} \left[ \epsilon_0 + \frac{1}{\beta} \log \mathop{\mathbb{E}}_{a'^{-n}} \left[ e^{\beta \tilde{Q}_2(s', a')} \right] - \frac{1}{\alpha} \log \pi^n(a'^n|s') \right] \right] \\
&= \gamma \epsilon_0 + r(s, a) + \gamma \mathop{\mathbb{E}}_{s'} \left[ \frac{1}{N} \sum_{n=1}^N \mathop{\mathbb{E}}_{a'^n \sim \pi^n} \left[ \frac{1}{\beta} \log \mathop{\mathbb{E}}_{a'^{-n}} \left[ e^{\beta \tilde{Q}_2(s', a')} \right] - \frac{1}{\alpha} \log \pi^n(a'^n|s') \right] \right] \\
&= \mathcal{T}_{\pi,\zeta}\tilde{Q}_2(s, a) + \gamma \epsilon_0
\end{aligned}
$$

Similarly, $\mathcal{T}_{\pi,\zeta}\tilde{Q}_1(s, a) \ge \mathcal{T}_{\pi,\zeta}\tilde{Q}_2(s, a) - \gamma \epsilon_0$

Therefore, $\left\| \mathcal{T}_{\pi,\zeta}\tilde{Q}_1 - \mathcal{T}_{\pi,\zeta}\tilde{Q}_2 \right\|_\infty \le \gamma \left\| \tilde{Q}_1 - \tilde{Q}_2 \right\|_\infty$, so $\mathcal{T}_{\pi,\zeta}$ is a $\gamma$-contraction. $\qquad\square$

**Theorem 1.** *(Convergence of MAOSVI)*

$\tilde{Q}_T$ *converges to* $\tilde{Q}^*$ *as* $T \to \infty$.

*Proof.* Let $\hat{\pi}_t^n(a^n|s) = \frac{e^{\alpha \hat{Q}_t^n(s,a^n)}}{\sum_{a'^n} e^{\alpha \hat{Q}_t^n(s,a'^n)}}$. Define $\mathcal{T}_\zeta \tilde{Q}(s,a) = \max_\pi \mathcal{T}_{\pi,\zeta} \tilde{Q}(s,a)$.

$$\arg\max_\pi \mathcal{T}_{\pi,\zeta} \tilde{Q}(s,a)$$

$$= \arg\max_\pi \left( r(s,a) + \gamma \mathbb{E}_{s'} \left[ \frac{1}{N} \sum_{n=1}^N \mathbb{E}_{a'^n \sim \pi^n} \left[ \frac{1}{\beta} \log \mathbb{E}_{a'^{-n} \sim \zeta^{-n}} \left[ e^{\beta \bar{Q}(s',a')} \right] - \frac{1}{\alpha} \log \pi^n(a'^n|s') \right] \right] \right)$$

$$= \arg\max_{\pi(\cdot|s)} \frac{1}{N} \sum_{n=1}^N \mathbb{E}_{a^n \sim \pi^n} \left[ \frac{1}{\beta} \log \mathbb{E}_{a^{-n} \sim \zeta^{-n}} \left[ e^{\beta \tilde{Q}(s,a)} \right] - \frac{1}{\alpha} \log \pi^n(a^n|s) \right]$$

$$= \hat{\pi}_t \text{ (According to \textbf{Lemma 1})}$$

Therefore, $\mathcal{T}_\zeta \tilde{Q}_t = \mathcal{T}_{\hat{\pi}_t, \zeta} \tilde{Q}_t$.

Assume that $\left| \tilde{Q}_1(s,a) - \tilde{Q}_2(s,a) \right| = \epsilon_0, \forall s, a$. According to **Lemma 2**, $\mathcal{T}_{\pi,\zeta}$ is a $\gamma$-contraction, hence

$$\mathcal{T}_{\pi,\zeta} \tilde{Q}_1(s,a) \le \mathcal{T}_{\pi,\zeta} \tilde{Q}_2(s,a) + \gamma \epsilon_0$$

Taking the maximum for both sides, we have

$$\max_\pi \mathcal{T}_{\pi,\zeta} \tilde{Q}_1(s,a) \le \max_\pi \left( \mathcal{T}_{\pi,\zeta} \tilde{Q}_2(s,a) + \gamma \epsilon_0 \right)$$

$$= \max_\pi \mathcal{T}_{\pi,\zeta} \tilde{Q}_2(s,a) + \gamma \epsilon_0$$

$$\text{i.e. } \mathcal{T}_\zeta \tilde{Q}_1(s,a) \le \mathcal{T}_\zeta \tilde{Q}_2(s,a) + \gamma \epsilon_0$$

Similarly, we have $\mathcal{T}_\zeta \tilde{Q}_1(s,a) \ge \mathcal{T}_\zeta \tilde{Q}_2(s,a) - \gamma \epsilon_0$, so $\mathcal{T}_\zeta$ is a $\gamma$-contraction, and hence $\tilde{Q}_T$ converges to the unique solution of Bellman equation $\mathcal{T}_\zeta \tilde{Q} = \tilde{Q}$ as $T \to \infty$. Further, since $\hat{\pi}_t^{1:N}$ is uniquely determined by $\tilde{Q}_t$ and continuous w.r.t $\tilde{Q}_t$, $\hat{\pi}_T^{1:N}$ also converges to the unique fix point of the iteration as $T \to \infty$.

For any policy $\pi$, if there exists $\pi'$ such that $\mathcal{T}_{\pi',\zeta} \tilde{Q}^\pi > \tilde{Q}^\pi$, then by applying $\mathcal{T}_{\pi',\zeta}^k$ to both sides we have $\mathcal{T}_{\pi',\zeta}^{k+1} \tilde{Q}^\pi > \mathcal{T}_{\pi',\zeta}^k \tilde{Q}^\pi$, and hence $\mathcal{T}_{\pi',\zeta}^k \tilde{Q}^\pi > \tilde{Q}^\pi$. Let $k \to \infty$ we have $\tilde{Q}^{\pi'} > \tilde{Q}^\pi$, so $\pi$ is not optimal. Thus, the optimal policy must be the unique fix point of the iteration to which $\hat{\pi}_T$ converges as $T \to \infty$, and hence $\tilde{Q}_T$ converges to $\tilde{Q}^*$ as $T \to \infty$. $\square$

**Theorem 2.** *(Near Optimality of Optimistic Soft Value Functions)*

1. *Assume that* $\pi^*_{(+\infty,\beta,\zeta)}$ *is unique, and let* $z = \min_{s \in \mathcal{S}} \zeta \left( a^*_{(+\infty,\beta,\zeta)}(s) \mid s \right)$, *then*
$$\left\| \tilde{Q}^*_{(+\infty,+\infty)} - \tilde{Q}^*_{(\alpha,\beta,\zeta)} \right\|_\infty \le \left( \frac{\log |\mathcal{A}|}{N\alpha} + \frac{\log \frac{1}{z}}{\beta} \right) \frac{\gamma}{1-\gamma} = O\left( \frac{1}{\alpha} + \frac{\log \frac{1}{z}}{\beta} \right).$$

2. *Let* $\Delta = \min_{s \in \mathcal{S}} \left( \tilde{Q}^*_{(+\infty,+\infty)}(s, a^*_{(+\infty,+\infty)}(s)) - \max_{a \notin a^*_{(+\infty,+\infty)}(s)} \tilde{Q}^*_{(+\infty,+\infty)}(s,a) \right)$,
   *and* $\Delta' = \min_{s \in \mathcal{S}} \left( \tilde{Q}^*_{(+\infty,\beta,\zeta)}(s, a^*_{(+\infty,\beta,\zeta)}(s)) - \max_{a \neq a^*_{(+\infty,\beta,\zeta)}(s)} \tilde{Q}^*_{(+\infty,\beta,\zeta)}(s,a) \right)$.
   *If* $\alpha > \frac{2\gamma \log |\mathcal{A}|}{\Delta'(1-\gamma)}$, $\beta > \frac{(1+\gamma) \log \frac{1}{z}}{(1-\gamma)\Delta}$, *then we have* $\left\| \tilde{Q}^{\pi^*_{(\alpha,\beta,\zeta)}}_{(+\infty,+\infty)} - \tilde{Q}^*_{(+\infty,+\infty)} \right\|_\infty \le$
   $$\left( \frac{|\mathcal{A}|^{\frac{2\gamma}{(1-\gamma)N}} \sum_n |\mathcal{A}^n|}{(1-\gamma) e^{\alpha \Delta'}} + \frac{2 \log |\mathcal{A}|}{N\alpha} + \frac{\log \frac{1}{z}}{\beta} \right) \frac{\gamma}{1-\gamma} = O\left( \frac{1}{\alpha} + \frac{\log \frac{1}{z}}{\beta} \right).$$

*Proof.*

1  According to the triangle inequality,

$$\left\|\tilde{Q}^*_{(+\infty,+\infty)} - \tilde{Q}^*_{(\alpha,\beta,\varsigma)}\right\|_\infty \le \left\|\tilde{Q}^*_{(+\infty,\beta,\varsigma)} - \tilde{Q}^*_{(\alpha,\beta,\varsigma)}\right\|_\infty + \left\|\tilde{Q}^*_{(+\infty,+\infty)} - \tilde{Q}^*_{(+\infty,\beta,\varsigma)}\right\|_\infty$$

$$\left\|\tilde{Q}^*_{(+\infty,\beta,\varsigma)} - \tilde{Q}^*_{(\alpha,\beta,\varsigma)}\right\|_\infty = \left\|\mathcal{T}^\infty_{+\infty,\beta,\varsigma}\tilde{Q}^*_{(\alpha,\beta,\varsigma)} - \tilde{Q}^*_{(\alpha,\beta,\varsigma)}\right\|_\infty$$

$$\le \sum_{k=0}^\infty \left\|\mathcal{T}^{k+1}_{+\infty,\beta,\varsigma}\tilde{Q}^*_{(\alpha,\beta,\varsigma)} - \mathcal{T}^k_{+\infty,\beta,\varsigma}\tilde{Q}^*_{(\alpha,\beta,\varsigma)}\right\|_\infty$$

$$\le \sum_{k=0}^\infty \gamma^k \left\|\mathcal{T}_{+\infty,\beta,\varsigma}\tilde{Q}^*_{(\alpha,\beta,\varsigma)} - \tilde{Q}^*_{(\alpha,\beta,\varsigma)}\right\|_\infty$$

$$= \frac{1}{1-\gamma}\left\|\mathcal{T}_{+\infty,\beta,\varsigma}\tilde{Q}^*_{(\alpha,\beta,\varsigma)} - \mathcal{T}_{\alpha,\beta,\varsigma}\tilde{Q}^*_{(\alpha,\beta,\varsigma)}\right\|_\infty$$

$$\left\|\tilde{Q}^*_{(+\infty,+\infty)} - \tilde{Q}^*_{(+\infty,\beta,\varsigma)}\right\|_\infty = \left\|\mathcal{T}^\infty_{+\infty,+\infty}\tilde{Q}^*_{(+\infty,\beta,\varsigma)} - \tilde{Q}^*_{(+\infty,\beta,\varsigma)}\right\|_\infty$$

$$\le \sum_{k=0}^\infty \left\|\mathcal{T}^{k+1}_{+\infty,+\infty}\tilde{Q}^*_{(+\infty,\beta,\varsigma)} - \mathcal{T}^k_{+\infty,+\infty}\tilde{Q}^*_{(+\infty,\beta,\varsigma)}\right\|_\infty$$

$$\le \sum_{k=0}^\infty \gamma^k \left\|\mathcal{T}_{+\infty,+\infty}\tilde{Q}^*_{(+\infty,\beta,\varsigma)} - \tilde{Q}^*_{(+\infty,\beta,\varsigma)}\right\|_\infty$$

$$= \frac{1}{1-\gamma}\left\|\mathcal{T}_{+\infty,+\infty}\tilde{Q}^*_{(+\infty,\beta,\varsigma)} - \mathcal{T}_{+\infty,\beta,\varsigma}\tilde{Q}^*_{(+\infty,\beta,\varsigma)}\right\|_\infty$$

Note that

$$\left|\mathcal{T}_{+\infty,\beta,\varsigma}\tilde{Q}^*_{(\alpha,\beta,\varsigma)}(s,a) - \mathcal{T}_{\alpha,\beta,\varsigma}\tilde{Q}^*_{(\alpha,\beta,\varsigma)}(s,a)\right|$$

$$= \gamma \left|\mathbb{E}_{s'}\left[\frac{1}{N}\sum_{n=1}^N\left(\max_{a'^n}\hat{Q}^n_{(\alpha,\beta,\varsigma)}(s',a'^n) - \frac{1}{\alpha}\log\sum_{a'^n}e^{\alpha\hat{Q}^n_{(\alpha,\beta,\varsigma)}(s',a'^n)}\right)\right]\right|$$

$$\le \gamma\,\mathbb{E}_{s'}\left[\frac{1}{N}\sum_{n=1}^N\left(\frac{1}{\alpha}\log\left(|\mathcal{A}^n|e^{\alpha\max_{a'^n}\hat{Q}^n_{(\alpha,\beta,\varsigma)}(s',a'^n)}\right) - \max_{a'^n}\hat{Q}^n_{(\alpha,\beta,\varsigma)}(s',a'^n)\right)\right]$$

$$= \frac{\gamma\log|\mathcal{A}|}{N\alpha}$$

$$\left|\mathcal{T}_{+\infty,+\infty}\tilde{Q}^*_{(+\infty,\beta,\varsigma)}(s,a) - \mathcal{T}_{+\infty,\beta,\varsigma}\tilde{Q}^*_{(+\infty,\beta,\varsigma)}(s,a)\right|$$

$$= \gamma\left|\mathbb{E}_{s'}\left[\max_{a'}\tilde{Q}^*_{(+\infty,\beta,\varsigma)}(s',a') - \frac{1}{N}\sum_{n=1}^N\max_{a'^n}\frac{1}{\beta}\log\mathbb{E}_{a'^{-n}\sim\varsigma^{-n}}e^{\beta\tilde{Q}^*_{(+\infty,\beta,\varsigma)}(s',a')}\right]\right|$$

$$\le \gamma\,\mathbb{E}_{s'}\left[\max_{a'}\tilde{Q}^*_{(+\infty,\beta,\varsigma)}(s',a') - \frac{1}{\beta}\log ze^{\beta\max_{a'}\bar{Q}^*_{(+\infty,\beta,\varsigma)}(s',a')}\right]$$

$$= \frac{\gamma\log\frac{1}{z}}{\beta}$$

Hence, $\left\|\tilde{Q}^*_{(+\infty,+\infty)} - \tilde{Q}^*_{(\alpha,\beta,\varsigma)}\right\|_\infty \le \left(\frac{\log|\mathcal{A}|}{N\alpha} + \frac{\log\frac{1}{z}}{\beta}\right)\frac{\gamma}{1-\gamma} = O\left(\frac{1}{\alpha} + \frac{\log\frac{1}{z}}{\beta}\right)$.

2  Since $\left\|\tilde{Q}^*_{(+\infty,+\infty)} - \tilde{Q}^*_{(+\infty,\beta,\varsigma)}\right\|_\infty \le \frac{\gamma\log\frac{1}{z}}{(1-\gamma)\beta}$, the gap of $\tilde{Q}_{(+\infty,\beta,\varsigma)}$ between the optimal joint action and any sub-optimal joint action is at least $\Delta - \frac{2\gamma\log\frac{1}{z}}{(1-\gamma)\beta}$, and for any agent, the gap of $\hat{Q}^n_{(+\infty,\beta,\varsigma)}$ between the optimal local action and any sub-optimal local action is at least $\Delta - \frac{2\gamma\log\frac{1}{z}}{(1-\gamma)\beta} - \frac{\log\frac{1}{z}}{\beta} = \Delta - \frac{(1+\gamma)\log\frac{1}{z}}{(1-\gamma)\beta}$. Therefore, when $\beta > \frac{(1+\gamma)\log\frac{1}{z}}{(1-\gamma)\Delta}$, $\pi^*_{(+\infty,\beta,\varsigma)}$ is also an optimal policy in the sense of the original value functions.

Further, since $\left\|\hat{Q}^*_{(+\infty,\beta,\zeta)} - \hat{Q}^*_{(\alpha,\beta,\zeta)}\right\|_\infty \leq \left\|\tilde{Q}^*_{(+\infty,\beta,\zeta)} - \tilde{Q}^*_{(\alpha,\beta,\zeta)}\right\|_\infty \leq \frac{\gamma \log |\mathcal{A}|}{(1-\gamma)N\alpha}$, the gap between an optimal local action and any sub-optimal local action is at least $d := \Delta' - \frac{2\gamma \log |\mathcal{A}|}{(1-\gamma)N\alpha}$. Therefore, following $\pi^*_{(\alpha,\beta,\zeta)}$, the probability that agent $n$ chooses the optimal local action is at least $\frac{e^{\alpha d}}{e^{\alpha d}+|\mathcal{A}^n|-1}$, and the probability of not choosing the optimal joint action is bounded by $\sum_n \left(1 - \frac{e^{\alpha d}}{e^{\alpha d}+|\mathcal{A}^n|-1}\right) \leq \sum_n \frac{|\mathcal{A}^n|}{e^{\alpha d}} = \frac{|\mathcal{A}|^{\frac{2\gamma}{(1-\gamma)N}} \sum_n |\mathcal{A}^n|}{e^{\alpha \Delta'}}$.

According to the triangle inequality,

$$\left\|\tilde{Q}^{\pi^*_{(\alpha,\beta,\zeta)}}_{(+\infty,+\infty)} - \tilde{Q}^*_{(+\infty,+\infty)}\right\|_\infty \leq \left\|\tilde{Q}^{\pi^*_{(\alpha,\beta,\zeta)}}_{(+\infty,+\infty)} - \tilde{Q}^*_{(\alpha,\beta,\zeta)}\right\|_\infty + \left\|\tilde{Q}^*_{(\alpha,\beta,\zeta)} - \tilde{Q}^*_{(+\infty,+\infty)}\right\|_\infty$$

$$\left\|\tilde{Q}^{\pi^*_{(\alpha,\beta,\zeta)}}_{(+\infty,+\infty)} - \tilde{Q}^*_{(\alpha,\beta,\zeta)}\right\|_\infty = \left\|\mathcal{T}^\infty_{+\infty,+\infty,\pi^*}\tilde{Q}^*_{(\alpha,\beta,\zeta)} - \tilde{Q}^*_{(\alpha,\beta,\zeta)}\right\|_\infty$$

$$= \sum_{k=0}^\infty \left\|\mathcal{T}^{k+1}_{+\infty,+\infty,\pi^*}\tilde{Q}^*_{(\alpha,\beta,\zeta)} - \mathcal{T}^k_{+\infty,+\infty,\pi^*}\tilde{Q}^*_{(\alpha,\beta,\zeta)}\right\|_\infty$$

$$\leq \sum_{k=0}^\infty \gamma^k \left\|\mathcal{T}_{+\infty,+\infty,\pi^*}\tilde{Q}^*_{(\alpha,\beta,\zeta)} - \tilde{Q}^*_{(\alpha,\beta,\zeta)}\right\|_\infty$$

$$= \frac{1}{1-\gamma}\left\|\mathcal{T}_{+\infty,+\infty,\pi^*}\tilde{Q}^*_{(\alpha,\beta,\zeta)} - \mathcal{T}_{\alpha,\beta,\pi^*,\zeta}\tilde{Q}^*_{(\alpha,\beta,\zeta)}\right\|_\infty$$

Note that

$$\left|\mathcal{T}_{+\infty,+\infty,\pi^*}\tilde{Q}^*_{(\alpha,\beta,\zeta)}(s,a) - \mathcal{T}_{\alpha,\beta,\pi^*,\zeta}\tilde{Q}^*_{(\alpha,\beta,\zeta)}(s,a)\right|$$

$$= \gamma\left|\mathbb{E}_{s'}\left[\mathbb{E}_{a'\sim\pi^*}\left[\tilde{Q}^*_{(\alpha,\beta,\zeta)}(s',a')\right] - \frac{1}{N}\sum_{n=1}^N \mathbb{E}_{a'^n\sim\pi^{*n}}\left[\hat{Q}^{*n}_{(\alpha,\beta,\zeta)}(s',a'^n) - \frac{1}{\alpha}\log \pi^{*n}(a'^n|s')\right]\right]\right|$$

$$\leq \gamma\mathbb{E}_{s'}\left[\max_{a'}\tilde{Q}^*_{(\alpha,\beta,\zeta)}(s',a') - \mathbb{E}_{a'\sim\pi^*}\left[\tilde{Q}^*_{(\alpha,\beta,\zeta)}(s',a')\right] - \frac{1}{N}\sum_{n=1}^N \mathbb{E}_{a'^n\sim\pi^{*n}}\left[\frac{1}{\alpha}\log \pi^{*n}(a'^n|s')\right]\right]$$

$$\leq \frac{\gamma|\mathcal{A}|^{\frac{2\gamma}{(1-\gamma)N}}\sum_n |\mathcal{A}^n|}{(1-\gamma)e^{\alpha\Delta'}} + \frac{\gamma\log|\mathcal{A}|}{N\alpha}$$

Combined with 1, we have

$$\left\|\tilde{Q}^{\pi^*_{(\alpha,\beta,\zeta)}}_{(+\infty,+\infty)} - \tilde{Q}^*_{(+\infty,+\infty)}\right\|_\infty \leq \left(\frac{|\mathcal{A}|^{\frac{2\gamma}{(1-\gamma)N}}\sum_n |\mathcal{A}^n|}{(1-\gamma)e^{\alpha\Delta'}} + \frac{2\log|\mathcal{A}|}{N\alpha} + \frac{\log\frac{1}{z}}{\beta}\right)\frac{\gamma}{1-\gamma}$$

$$= O\left(\frac{1}{\alpha} + \frac{\log\frac{1}{z}}{\beta}\right)$$

$\square$

**Theorem 3.** *(Convergence of MAOSQL) If $\beta > \frac{2(N-1)\alpha}{1-\gamma}$, then $\hat{\pi}^{1:N}_T$ converges to $\pi^{1:N}_*$ as $T \to \infty$, such that $\pi^{1:N}_*$ optimizes $\tilde{Q}^{\pi^*}$, the unique solution of Bellman equation $\mathcal{T}_{\pi,\zeta}\tilde{Q} = \tilde{Q}$ for some $\zeta$.*

*Proof.* Define the Bellman operator on $\hat{Q}$ as

$$\hat{\mathcal{T}}\hat{Q}^n(s,a^n) = \frac{1}{\beta}\log\mathbb{E}_{a^{-n}\sim\hat{\zeta}^{-n}}\left[\exp\left(\beta\left(r(s,a) + \gamma\mathbb{E}_{s'}\left[\frac{1}{N}\sum_{n'=1}^N \hat{V}^{n'}(s')\right]\right)\right)\right] \quad \text{where}$$

$$\hat{\zeta}(a|s) = \frac{\epsilon}{|\mathcal{A}|} + (1-\epsilon)\hat{\pi}(a|s)$$

$$\hat{\pi}^n(a^n|s) = \exp\left(\alpha\left(\hat{Q}^n(s,a^n) - \hat{V}^n(s)\right)\right)$$

$$\hat{V}^n(s) = \frac{1}{\alpha}\log\sum_{a^n}e^{\alpha\hat{Q}^n(s,a^n)}$$

Assume that $\left\|\hat{Q}_1 - \hat{Q}_2\right\|_\infty = \epsilon_0$, so $\hat{Q}_1^n(s, a^n) \leq \hat{Q}_2^n(s, a^n) + \epsilon_0, \forall n, s, a^n$

$$\text{then } \hat{V}_1^n(s) = \frac{1}{\alpha} \log \sum_{a^n} e^{\alpha \hat{Q}_1^n(s, a^n)}$$

$$\leq \frac{1}{\alpha} \log \sum_{a^n} e^{\alpha\left(\hat{Q}_2^n(s, a^n) + \epsilon_0\right)}$$

$$= \hat{V}_2^n(s) + \epsilon_0$$

Similarly, $\hat{V}_1^n(s) \geq \hat{V}_2^n(s) - \epsilon_0$, so $\left\|\hat{V}_1 - \hat{V}^2\right\|_\infty \leq \epsilon_0$.

$$\left|\hat{\mathcal{T}}\hat{Q}_1^n(s, a^n) - \hat{\mathcal{T}}\hat{Q}_2^n(s, a^n)\right| = \frac{1}{\beta} \log \sum_{a^{-n}} \left[\exp\left(\left|\log \frac{\hat{\zeta}_1^{-n}(a^{-n}|s)}{\hat{\zeta}_2^{-n}(a^{-n}|s)}\right| + \beta\gamma\mathbb{E}_{s'}\left[\frac{1}{N}\sum_{n'=1}^{N}\left|\hat{V}_1^{n'}(s') - \hat{V}_2^{n'}(s')\right|\right]\right)\right]$$

$$\left|\log \frac{\hat{\zeta}_1^{-n}(a^{-n}|s)}{\hat{\zeta}_2^{-n}(a^{-n}|s)}\right| = \left|\log \frac{\frac{\epsilon}{|\mathcal{A}^{-n}|} + (1-\epsilon)\hat{\pi}_1^{-n}(a^{-n}|s)}{\frac{\epsilon}{|\mathcal{A}^{-n}|} + (1-\epsilon)\hat{\pi}_2^{-n}(a^{-n}|s)}\right|$$

$$\leq \left|\log \frac{\hat{\pi}_1^{-n}(a^{-n}|s)}{\hat{\pi}_2^{-n}(a^{-n}|s)}\right|$$

$$= \left|\alpha \sum_{n' \neq n}\left(\hat{Q}_1^{n'}(s, a^{n'}) - \hat{V}_1^{n'}(s) - \hat{Q}_2^{n'}(s, a^{n'}) + \hat{V}_2^{n'}(s)\right)\right|$$

$$\leq 2(N-1)\alpha\epsilon_0$$

Therefore,

$$\left|\hat{\mathcal{T}}\hat{Q}_1^n(s, a^n) - \hat{\mathcal{T}}\hat{Q}_2^n(s, a^n)\right| \leq \left(\frac{2(N-1)\alpha}{\beta} + \gamma\right)\epsilon_0$$

Since $\beta > \frac{2(N-1)\alpha}{1-\gamma}$, $\frac{2(N-1)\alpha}{\beta} + \gamma < 1$, so $\hat{\mathcal{T}}$ is a $\left(\frac{2(N-1)\alpha}{\beta} + \gamma\right)$-contraction, $\hat{Q}_T^{1:N}$ converges to the unique fix point of Bellman equation $\hat{\mathcal{T}}\hat{Q} = \hat{Q}$ as $T \to \infty$. Further, since $\hat{\pi}$ is uniquely determined by $\hat{Q}$ and continuous w.r.t $\hat{Q}$, $\hat{\pi}_T^{1:N}$ also converges to the unique fix point of the iteration as $T \to \infty$.

Note that at convergence, $\hat{\zeta}_T$ is fixed to some $\zeta$, so the corresponding $\tilde{Q}$ is the fix point of the $\mathcal{T}_\zeta$. According to **Theorem 1**, $\tilde{Q}_T$ converges to $\tilde{Q}^*$. □

## C  IMPLEMENTATION DETAILS

Our implementation of MAOSDQN and all experiments are based on EPyMARL (Papoudakis et al. (2020)). To ensure fair comparison, we adopt the same buffer size, number of network parameters and optimizer for all algorithms. For the behavior policy of MAOSDQN, we adopt an epsilon-softmax policy with $\epsilon_b$ linearly anealing from $1.0$ to $0.05$ within $5000$ steps, which is the same as that in QMIX's $\epsilon$-greedy behavior policy.

In our early experiments, we found that Reward standardization, Double Q-learning and Dueling Q-network are essential to stable training, so our final implementation also include these three tricks.

There are 3 important hyper-parameters in MAOSDQN. In Tensor Game and Matrix Game, we set $\alpha = 50, \beta = 10, \epsilon = 0.05$, while in Level-based Foraging, we set $\alpha = 20, \beta = 1, \epsilon = 0.5$.

