# OpenReview forum: "Multi-agent Optimistic Soft Q-Learning: A co-MARL Algorithm with a Global Convergence Guarantee"
_ICLR.cc/2024/Conference — ICLR 2024 Conference Withdrawn Submission_

### Official Review · Reviewer_Vgbg · 2023-10-24

**Soundness:** 2 fair
**Presentation:** 2 fair
**Contribution:** 2 fair
**Rating:** 3
**Confidence:** 4

**Summary:**

The paper addresses two key issues in existing MARL methods: miscoordination and relative overgeneralization. To tackle these problems, the paper introduces local Q-functions as the logsumexp of global Q-functions with an energy-based policy. The authors then derive a multi-agent optimistic soft value iteration algorithm to learn nearly optimal policies and extend it to soft Q-learning using neural networks. Experimental results confirm the effectiveness of their proposed method.

**Strengths:**

The idea of using soft value functions to approximate optimality is interesting and novel.
The paper provides sufficient theoretical derivations and proofs.

**Weaknesses:**

Some notations require further clarification.
Experimental evaluations appear somewhat simplistic.
It is unclear how the proposed method's optimality relates to the original MARL problem.
Specific questions have been raised, as detailed below.

**Questions:**

1. The paper states that policy gradient methods always fail to converge to the globally optimal solution. Please clarify which policy gradient algorithm in MARL this statement refers to and provide supporting evidence for the claim. The original explaination is unclear.

2. The paper mentions that there is no theoretical guarantee that QTRAN and QPLEX can resolve relative overgeneralization, despite these methods being considered sufficient and necessary in IGM. Please provide further explanation if the authors think there is a gap between the optimality guarantee in this paper and their guarantees under IGM.

3. Where is $\hat{Q}^n$ defined? Is it $\hat{Q}^n_\zeta$?

4. The objective defined in Lemma 1 appears different from the original expected return in RL or an entropy-regularized one. Please provide further explanation regarding what it means to achieve optimality in the context of this paper's defined objective function.

5. The paper mentions that the proposed definition is equivalent to the "original" one when \alpha and \beta approach infinity. Please clarify what "original" means, along with the "original" problem mentioned in Theorem 2.

6. What does $\widetilde{Q}^*_{(\infty, \infty)}$ mean? While the fixed point of the original Bellman equation represents the optimal value function, please explain the significance of the fixed point of the Bellman operator defined in this paper.

7. Where is $a^*_{(\alpha,\beta,\gamma)}$ defined?

8. The paper mentions that assigning different $\zeta(·|s)$ on symmetric actions can break the symmetry. Given that miscoordination is one of the problems the paper aims to solve, please provide detailed explanations on how this mechanism addresses the issue, and how it works practically.

9. What is the behavior policy in Algorithm 2 and 3? Is it $\hat{\zeta}$?

10. The $L_{extra}$ term needs further explanation. Is it necessary for the experimental results? What does it signify when $\hat{Q}^n(s,a^{\*n}) > \widetilde{Q}(s,a^*)$?

11. Why not compare the proposed method with other value-based methods like QTRAN and QPLEX in experiments, considering that the proposed method is also value-based?

12. The experiments appear overly simplistic. It would be beneficial to demonstrate the performance of the proposed method in more complex domains such as SMAC and GRF.

13. The LBF experiments do not effectively demonstrate the proposed method's superiority. MAOSDQN does not outperform value-based QMIX or other policy-based methods. Additionally, please explain why $\epsilon$ is set to be 0.5, which differs from QMIX in LBF.

14. The proposed method seems highly sensitive to the parameters $\alpha$ and $\beta$. It is recommended that the authors conduct ablation studies to illustrate how these parameters affect optimality and convergence.

15. While the paper proves that a large $\beta$ can help in finding optima, one might suggest that a reward shaping method, e.g., $r' = e^{\beta r}$, with a large $\beta$, could yield similar results.

---

### Official Review · Reviewer_8w13 · 2023-10-30

**Soundness:** 2 fair
**Presentation:** 2 fair
**Contribution:** 2 fair
**Rating:** 3
**Confidence:** 4

**Summary:**

This paper aims to gain new theoretical understandings of multi-agent reinforcement learning algorithms in the realm of centralized training with decentralized execution. The authors found that current algorithms struggle with some basic tasks because of miscoordination and relative overgeneralization. To solve this, they introduce a new method called "multi-agent optimistic soft q-learning", which is implemented with an optimistic local Q-function and a softmax local policy. Here are the key points about this new solution:

1.	It is a solution with a provable global convergence guarantee.
2.	Its objective gives near-optimal policies with a tractable error bound.
3.	It can be easily modified for deep reinforcement learning scenarios.

To test this model, they tried it in simple learning environments as well as more complex ones, like the level-based foraging environment.

**Strengths:**

1. The subject of this research is quite interesting. It focuses on addressing two major problems, which are miscoordination and relative overgeneralization, all within a single approach.

2. I really appreciate how the authors included clear examples in Section 2.3. These examples make it easier to grasp the concepts of these two issues.

3. The authors have put in a lot of work to back up their solution with solid theoretical evidence.

**Weaknesses:**

1. It's not entirely clear if the problems of miscoordination and relative overgeneralization are really significant in practical uses of multi-agent reinforcement learning. This uncertainty makes it difficult to assess the true impact of this research.

2. The experiments conducted seem quite basic, which doesn't provide enough evidence to fully support the authors' claims. For instance, in Sections 4.1 and 4.2, it would be helpful if the authors offered deeper insights into their experimental results, explaining why other methods didn't work and what makes their proposed solution effective in these scenarios.

3. The choice of baseline models for comparison seems improper. There are several existing studies that address the issues mentioned. For example, regarding the problem of relative overgeneralization, the authors could have referred to works like "Lenient Multi-Agent Deep Reinforcement Learning" by Gregory Palmer et al., "Actor-Attention-Critic for Multi-Agent Reinforcement Learning" by Shariq Iqbal and Fei Sha, and "Probabilistic Recursive Reasoning for Multi-Agent Reinforcement Learning" by Ying Wen et al. These studies would have been more appropriate for comparison.

Some small points:

1. When talking about “optimal adaptive learning (OAL) (Wang & Sandholm (2002)) is the first algorithm with provable convergence to the optimal policy”, it might be useful to also mention the earlier work "Nash convergence of gradient dynamics in general-sum games" by Satinder Singh, Michael Kearns, and Yishay Mansour, which also discusses the convergence to Nash equilibrium.

2. The text size in Figures 1 to 3 could be made bigger for easier reading.

**Questions:**

1. How serious are the problems of miscoordination and relative overgeneralization in real-life situations?

2. Can you offer more explanation about why earlier studies couldn't tackle these two issues effectively?

3. In the section on related work, you mentioned that previous research mainly concentrated on achieving Nash equilibrium in general-sum games and didn't focus much on cooperative settings. Could you elaborate on the differences between cooperative games and general-sum games? Are solutions designed for general-sum games still applicable in cooperative scenarios?

---

### Official Review · Reviewer_njtx · 2023-11-01

**Soundness:** 2 fair
**Presentation:** 3 good
**Contribution:** 2 fair
**Rating:** 5
**Confidence:** 3

**Summary:**

This paper presents a novel multi-agent RL algorithm, which involves centralized training and distributed execution. Specifically, the authors design algorithms including MAOSQL, MAOSVI, and MAOSDQN. These algorithms are different variants and share the similar underlying convergence behavior. The authors theoretically analyze the global convergence and justify their claims. To validate the proposed algorithms, the authors use three benchmark models and a few baselines to show the superiority.

**Strengths:**

The investigated topic in this paper is quite interesting and critical and relevant to the community. The paper is well motivated and seems theoretically strong.

**Weaknesses:**

1. The authors mentioned two critical issues: miscoordination and overgeneralization. However, in the convergence guarantee, it is unclear to me how the authors have addressed these two issues.

2. For these three algorithms, the selection of hyper-parameters are extremely important. Are there any specific rules for them to choose for different scenarios? Or they follow the similar values in various cases? In Theorem 2, the error bounds are related to these parameters. What are the values of them such that the bounds are tight?

3. The epsilon-decentralized-softmax policy requires the $\epsilon$, how to determine this value in practical scenarios?

4. The experimental results are not promising. All games need variances in the plot or tables. Also, the information in Tables 4 and 5 is confusing. Additionally, the results in Table 6 did not show the superiority of the proposed MAOSDQN as other algorithms outperform in various tasks.

5. Please clarify the notations in all theoretical statements to make them more readable.

**Questions:**

Please see the questions in the above weaknesses.

---

### Official Review · Reviewer_aokn · 2023-11-09

**Soundness:** 2 fair
**Presentation:** 3 good
**Contribution:** 2 fair
**Rating:** 6
**Confidence:** 4

**Summary:**

The paper presents a co-MARL algorithm that solves miscoordination and relative overgeneration problems. The algorithm defines a local value function and a relation between the local value function and local policy of each agent. The proposed approach was also extended to neural network function approximators.

**Strengths:**

The paper is well-written and the empirical results shows that the proposed algorithm with neural network approximators, MAOSDQN, performs better than the baselines.

**Weaknesses:**

In the empirical results, I am not convinced by the discussion for why the algorithms (including MAOSDQN) behave like that. More importantly, it seems that QMIX does not learn in the single-state Matrix game. I would appreciate if the authors can expand upon the discussion in section 4 and explain why we see a significant performance improvement between MAOSDQN and the baselines.

**Questions:**

I don’t have questions at this revision cycle.